# Eating Habits Associated with Nutrition-Related Knowledge among University Students Enrolled in Academic Programs Related to Nutrition and Culinary Arts in Puerto Rico

**DOI:** 10.3390/nu12051408

**Published:** 2020-05-14

**Authors:** Christian Rivera Medina, Mercedes Briones Urbano, Aixa de Jesús Espinosa, Ángel Toledo López

**Affiliations:** 1Jose A Tony Santana International School of Hospitality & Culinary Arts, Universidad Ana G Méndez, San Juan 00901, Puerto Rico; 2Health and Nutrition, International Iberoamerican University, 902114799 Barcelona, Spain; mercedes.briones@unini.org; 3Neurosurgery Section, University of Puerto Rico Medical Science Building, San Juan 00921, Puerto Rico; aixa.dejesus@upr.edu; 4Social and Human Sciences, Universidad Ana G Méndez, San Juan 00921, Puerto Rico; atoledo@uagm.edu

**Keywords:** university, student, nutrition, knowledge, culinary arts, eating habits, nutritional requirements

## Abstract

University students frequently develop unhealthy eating habits. However, it is unknown if students enrolled in academic programs related to nutrition and culinary arts have healthier eating habits. We evaluated the relationship of eating habits and nutritional status of students in academic programs with knowledge on nutrition, as well as cooking methods and techniques. A descriptive cross-sectional study was conducted in spring of 2019, while we completed a survey measuring eating habits and knowledge on nutrition, as well as cooking methods and techniques. Anthropometric measurements were collected for nutritional status estimation. The non-probabilistic convenience sample comprised 93 students pursuing degrees at Universidad Ana G. Mendez, Puerto Rico. Inadequate body mass index (BMI) was observed in 59% of the students. Eating habits, knowledge on nutrition, and knowledge on cooking methods and techniques were inadequate in 86%, 68%, and 41% of the population, respectively. Eating habits were associated with knowledge on nutrition and academic program, but not with knowledge on cooking methods and techniques. Most students reported having inadequate eating habits and BMI. Nutrition and dietetics students had the best knowledge on nutrition compared to culinary management students, a majority of whom had inadequate knowledge. We can conclude that there are other factors inherent to students’ life that may have a stronger influence on eating habits.

## 1. Introduction

Eating habits are defined as “conscious, collective, and repetitive behaviors, which lead people to select, consume, and use certain foods or diets, in response to social and cultural influences” [1]. University students are in a stage of change that renders them more susceptible to the development of unhealthy eating habits. Stress, short sleep durations, economic limitations, lack of time, and lifestyle-related changes are some factors that affect eating habits [2].

Given their field of study and nature of coursework, university students enrolled in programs related to health and food, such as nutrition and dietetics, culinary nutrition, and culinary management, are expected to have better eating habits than those of their peers. The existing literature presents conflicting results pertaining to whether knowledge on nutrition, as well as cooking methods and techniques, may influence the eating habits of university students [3,4,5,6]. It is unknown if knowledge on nutrition and cooking methods and techniques within the Puerto Rican student population represents a protection factor that leads them to practice healthy eating habits or if, on the contrary, these students present eating habits that are similar to those of the rest of the university population.

Mihalopoulos et al. [7] stated that, once students are exposed to university life, they tend to gain 15 pounds as they acquire unhealthy eating habits that promote the development of overweight and obesity. Additionally, Sanchez-Ojeda et al. [8] demonstrated that students enrolled in health-related academic programs have unhealthy lifestyles, characterized by unbalanced and deficient diets, presenting overweight and obesity.

It is important to investigate the eating habits of Puerto Rican university students, whose diets often feature low dietary fiber intake, and higher intakes of carbohydrates such as rice, bread, cereal, and refined flour-based foods. Such students also show high intakes of dairy products such as whole milk and its derivatives, refined sugar, legumes, tubers, and a variety of animal proteins with a high fat content, causing their diets to be high in calories, simple carbohydrates, saturated fats, and sodium [9,10]. In addition, they tend to consume fast food, sweets, carbonated drinks, and fried foods due to their accessibility and affordability [11,12].

The Puerto Rico Department of Health in its report, Puerto Rico Community Health Assessment: Secondary Data Profile [13], indicated that the prevalence of overweight and obesity increased since 1996. Puerto Rico Behavioral Risk Factor Surveillance System statistics show that 66% of the population had overweight and obesity [14], significantly increasing the prevalence of chronic diseases such as cardiovascular disease and diabetes, which are invariably associated with eating habits [15].

Wilson et al. [16] identified that university students have inadequate cooking skills, rendering them susceptible to the development of poor eating habits that promote weight gain. Having a high level of knowledge on cooking methods and techniques at the time of food preparation may result in lower intake levels of processed foods and higher intakes of foods prepared from scratch [17,18]. Scientific evidence links university students with knowledge on cooking methods and techniques with better diet quality [19], a higher consumption of vegetables and fruits, and a more varied and balanced dietary intake [20]. Furthermore, positive influences on body mass index (BMI) and better overall health were also observed in these students [20]. The benefits of self-confidence and knowledge on food preparation were also evidenced in a study by Reicks et al. [21], who found that knowledge on food preparation increased the intake of fruits and vegetables.

Students enrolled in nutrition-related programs are expected to be at an advantage in terms of the acquisition and learning of nutritional concepts in comparison to other fields of study and in a better position to implement healthy eating habits, as their academic program provides competencies for this purpose. Therefore, it is expected that such students can implement the information received in the course of their study to improve their eating habits and lifestyles. However, a study conducted among university students at the University of Alicante Spain Department of Human Nutrition and Dietetics by Rizo-Baeza et al. [22] demonstrated that the consumption of macro and micronutrients did not adhere to dietary guidelines, showing that the presence of nutrition-related knowledge does not influence decision-making pertaining to healthy diets and lifestyles.

Accordingly, this study aimed to evaluate the relationship of the eating habits and nutritional status of university students enrolled in academic programs related to nutrition and culinary arts with their knowledge on nutrition, as well as cooking methods and techniques.

## 2. Materials and Methods

### 2.1. Study Design

This cross-sectional study focused on the eating habits, nutritional status, knowledge levels pertaining to nutrition, and the cooking methods and techniques of university students. For eligibility to participate, students had to be at least 18 years or older and pursuing a bachelor’s degree in streams related to nutrition and culinary management at the Universidad Ana G. Mendez. All subjects gave their informed consent for inclusion before they participated in the study. The study was conducted in accordance with the Declaration of Helsinki, and the protocol was approved by the Ethics Committee of Universidad Ana G Mendez (06-085-18).

After study approval, the investigator coordinated with faculty members based on the academic calendar dates and time for the administration of the questionnaire within their classrooms. Once the students were present in the classrooms, the researcher provided information to all prospective participants regarding the study and its objectives, what participation would entail, including the measurement of weight and height for BMI estimation, and the length and duration of the self-administered questionnaire. Informed consent was provided by these students, with the information discussed by the researcher. Prospective participants were informed that no incentive for participation would be offered, and there were no penalties for discontinuing participation. Each student could drop out of the study at any time during the administration of the questionnaire by hand. Following this, those interested in participating voluntarily were provided the questionnaire. The researcher was present at all times during the administration of the questionnaire to clarify possible doubts and answer questions.

After students completed the self-administered questionnaire, the researcher obtained their weight and height values. An appropriate distance was established that separated participants whose values were being measured from the rest of the group to safeguard their privacy. These values were noted in the predetermined area of the questionnaire.

### 2.2. Survey Development

To develop an instrument capable of measuring students’ eating habits and knowledge on nutrition, as well as cooking methods and cooking techniques, several existing instruments were reviewed. A panel of experts, comprising an expert in evaluation processes, a chef, a nutritionist–dietitian licensed to practice in Puerto Rico, and a doctor in nutrition, was consulted to review the initial instrument. After this, a pilot test was conducted on 16 students, to evaluate the ease of use and clarity of the instrument. Based on the results of the pilot study, items were revised, merged, or eliminated. In addition, content was reworded as needed to achieve good comprehension. The survey was also approved by the Academic Commission of the International Iberoamerican University UNINI México. This process promoted the adaptation of queries to meet linguistic and cultural aspects related to the sample under investigation, and questions from previously validated surveys were identified and used to develop the current questionnaire.

The initial survey comprised 170 items, with the final version comprising 164 items, including closed multiple-choice questions and assessment Likert scales. The instrument was divided into four sections: student profile (socio-economic and personal characteristics), eating habits, knowledge on nutrition, and knowledge on cooking methods and cooking techniques.

The student profile section included questions regarding sociodemographic information such as age, sex, family composition, civil status, residence, income, and employment. In terms of education, students were asked about their year of study, career of choice, and if they were studying part-time or full time. Questions regarding the preparation of food in the residence, use of kitchen equipment and meal planning, caloric intake, type of diet, and physical activity were also included.

#### 2.2.1. Eating Habits

Specific questions regarding eating habits were used from the Youth Risk Behavior Surveillance System of the Center for Disease Control [23] for the identification of unhealthy dietary behaviors. Additionally, questions from the “NHSGGC Community-Based Cooking Skills Program Follow-Up Questionnaire” by García et al. [24] were also used since they assisted in the identification of barriers within cooking and healthy eating, and they allowed for the measurement of elements in food planning, purchasing, and food preparation from scratch. Specific questions from The Food Consumption Frequency Questionnaire by Goni et al. [25] were also integrated, allowing for the identification of nutritional alterations caused by an inadequate diet, and observation of the possible quantity and quality of the foods consumed during a certain period of time, as well as eating habits.

Students were asked about the frequency of consumption of foods and beverages in the last month to further measure their dietary diversity and to capture individual usual food consumption levels by obtaining data on the frequency with which the student consumed food items based on a predefined food list. Based on previous surveys [21,23,24], the list included 34 items within the basic food groups (fruits, vegetables, proteins, dairy products, grains, fats), ways of consuming foods (boiled, fried), and eating or cooking habits (eating out, cooking at home, ready-made meals, preparing from scratch). In addition, nine questions pertaining to food omission, food portions, and weight loss strategies, as well as elements considered in the nutritional facts label, were included to better identify and understand the application of healthy eating habits.

A section detailing the kinds of meals that were consumed during the day (breakfast, lunch, dinner, and snacks) and which food groups were included in each of the daily meals was employed. Students were asked about the application of various cooking methods and techniques (boiled, steam, stew, fried, sautéed, roasted, grill, bake, and sous-vide) on a provided list of proteins, vegetables, and carbohydrates. This set of inquiries facilitated the investigation of the effect of students’ application of cooking methods and techniques on their food consumption.

An evaluative scale was developed for eating habits. When students reported a correct health eating habit based on dietary guidelines [26], a score of 1 was assigned. Furthermore, if the student consumed proteins, vegetables, or carbohydrates in more than four cooking methods or techniques, one point was awarded. Contrarily, if the frequency of consumption, meals consumed during the day, and food groups included in each meal were not in accordance with nutritional guidelines, no points were given. An average of correct eating habits was estimated for each student. Values between 0% and 69% were considered “inadequate,” those between 70% and 79% considered “satisfactory,” and those between 80% and 100% considered “adequate.”

#### 2.2.2. Knowledge on Nutrition

To measure the level of knowledge on nutrition and whether it was adequate, satisfactory, or inadequate, 36 specific questions were adapted from a study by Tamayo et al. [27]. The questions acquired from this survey help in the recognition of the areas of limitation in students’ understanding of healthy eating habits. The questions included were based on the level of knowledge on nutritional recommendations regarding daily food and water intake, food groups in the plate, food portions, and benefits of fiber consumption. Furthermore, the ability to identify the risk associated with the intake of excess salt and drinks with high concentrations of sugar, healthy cooking techniques, and unhealthy fat intake, and yes/no questions about basic nutritional concepts pertaining to high-fiber foods, foods with cholesterol, nutritional fact labels, and healthy habits for portion control were also evaluated. Students were awarded one point when a correct answer was stated, while no points were allocated to incorrect answers, and an average of correct answers was estimated. Values between 0% and 69% were considered “inadequate,” those between 70% and 79% considered “satisfactory,” and those between 80% and 100% considered “adequate.”

#### 2.2.3. Knowledge on Cooking Methods and Techniques

The section on cooking methods and techniques comprised 55 questions divided into three subsections: confidence in culinary competencies, knowledge on methods and techniques, and frequency of the application of cooking methods and techniques. Data from a previous study of culinary skills and consumption of processed or prepared foods in university students [28], as well as a cooking skills scale used in a study by Hartmann et al. [29], were used for the development of this section. The selected sections of these surveys were included within this survey as they measured students’ cooking competencies and knowledge on how to cook, as well as their relationship with eating habits. Specific questions from the Basic Culinary Aptitude Questionnaire were also adapted, for the measurement of students’ knowledge levels and applicability of cooking methods and techniques [30]. The General Knowledge Questionnaire on Nutrition and Food, pertaining to food, nutrients, and alterations, or processes related to food, was used [31].

Finally, a questionnaire on cooking skills based on a study by Ternier [32] was used to determine if knowledge on and skills in food preparation influenced food purchases and consumption. This section included a self-confidence scale for cooking competencies and questions on the use of cooking methods and techniques. One point was assigned when the student answered “confident” or “very confident” with regard to culinary competencies. In addition, we also employed yes/no questions for the identification of cooking methods and techniques, as well as recommended techniques for the nutrient retention of certain foods. If the student was able to answer the statement correctly, one point was awarded. A Likert scale (0 denoting “never” and 3 denoting “always”) was also used to measure cooking frequency and practice. If the frequency of the application of culinary practices was “most of the time” or “all the time,” one point was awarded. Similar to the previous sections, values between 0% and 69% were considered “inadequate,” those between 70% and 79% considered “satisfactory,” and those between 80% and 100% considered “adequate.”

### 2.3. Statistical Analysis

Descriptive statistics such as frequency distributions, proportions, means, and standard deviations (SDs) were used to summarize the data. Associations between categorical variables were measured using Fisher’s exact test or a chi-square test. Statistical significance was set at *p* < 0.05. Data were analyzed using the Statistical Package for the Social Sciences (version 25, 2019 IBM SPSS Statistics) [33] and R (version 3.4.4) [34].

### 2.4. Sample

The sample of this study comprised all university students pursuing a bachelor’s degree at the Universidad Ana G. Méndez in Puerto Rico and who were enrolled in the following academic programs: nutrition and dietetics, culinary nutrition, and culinary management. At the time this investigation took place, there were only a total of 93 students enrolled dispersed within the three academic programs that complied with inclusion criteria. As the target population was small, non-probabilistic intentional sampling was used, wherein the total population of the students of interest was chosen at the time of questionnaire administration. A total of 93 students completed the survey.

## 3. Results

### 3.1. Participant Characteristics

Data on the student profile are shown in Table 1. The sample comprised 59 (63.4%) women, with an average age of 24.3 ± 6.7 years. The majority of the students were single (*n* = 69; 74%) and had a job (*n* = 62; 66%). Most of the students (*n* = 85; 91.4%) reported cooking food at home and having an omnivorous diet (*n* = 70; 75.2%). In terms of career, the program with the highest number of students was culinary management (*n* = 42, 45.2%), followed by culinary nutrition (*n* = 28, 30.1%). Only 12 students were in their first year, and there were no first-year students in the dietetics and nutrition program. A total of 23 students were in their second year, and 29 students each were in their third and fourth years, respectively. A majority (*n* = 73; 78.5%) of the students were classified as having a low income.

Fifty-five students (59.1%) had an inadequate BMI, presenting low weight (*n* = 5; 5.3%), overweight (*n* = 25; 26.9%), and obesity (*n* = 25; 26.9%). A statistically significant difference was observed between students’ BMI and academic program type. A higher percentage of students in the nutrition and dietetics program (56%) had a normal weight, followed by those in the culinary nutrition program (50%). The academic program with the highest prevalence of obesity was culinary management (*n* = 18; 42.9%), followed by culinary nutrition (*n* = 5, 17.9%).

### 3.2. Eating Habits

Table 2 shows the summary of the students’ eating habits. Students’ correct eating habits ranged from 27.8% to 82.9% with a mean of 58.3% (SD = 11.7%). A majority of the students (*n* = 80, 82.9%) had inadequate eating habits, followed by nine students with satisfactory habits, and four with adequate eating habits.

Adequate eating habits were reflected in household food consumption (*n* = 90; 96.8%) or cooking from scratch (*n* = 78; 83.9%) and dinner consumption every day (*n* = 76; 81%). The inadequate eating habits of the majority of the sample pertained to the intake of nutritionally unbalanced meals lacking the integration of the five food groups. As shown in Table 2, the students’ inadequate eating habits based on dietary guidelines were characterized by high intake levels of fried foods, charcuterie, and carbonated beverages. In addition, students were not adhering to the dietary guidelines regarding the consumption of vegetables, fruits, water, whole milk, and 100% fruit juice, as well as portion control and snacking two to three times daily.

### 3.3. Knowledge on Nutrition

The level of knowledge on nutrition varied from 22.1% to 91.7%, according to the evaluation scale, with the majority of students (*n* = 64, 68.8%) showing inadequate knowledge levels, followed by 16 (17.2%) with satisfactory levels, and 13 (14%) with adequate levels, as defined in this survey. Table 3 shows correct knowledge regarding questions related to nutrition by the majority of students. None of the first-year students had adequate nutrition-related knowledge levels. Of those with adequate levels, more than half (*n* = 7, 53.8%) were in their fourth year and 38.5% were in their third year. No association was observed between students’ knowledge on nutrition and their BMI (*p* = 0.46).

### 3.4. Knowledge on Cooking Methods and Technique

The level of knowledge on cooking methods and techniques was between adequate and satisfactory in 58% of the students. Students felt, on average, confident or very confident in 18 of the 26 culinary competencies included in this section of the questionnaire. Table 4 shows a summary of the premises on which the students presented adequate or inadequate knowledge levels. The mean score of the subsection measuring the level of knowledge on cooking methods and techniques was 61.4% ± 18%, with the lowest score being 0% and the highest being 95%. Culinary nutrition students presented the highest knowledge levels in terms of cooking methods and techniques (*n* = 18; 64.3%), followed by those enrolled in culinary management (*n* = 26; 61.9%). Fewer than half of the students in the nutrition and dietetics program (*n* = 10, 43.5%) reported adequate or satisfactory knowledge levels for cooking methods and techniques. No association was found between students’ knowledge on cooking methods and techniques and their eating habits (*p* = 0.35), university career (*p* = 0.61), or BMI (*p* = 0.54).

### 3.5. Bivariate Analysis

An association was observed between academic program type and eating habits, which revealed that culinary management students had the most inadequate eating habits. Similarly, knowledge on nutrition was associated with academic program type, with culinary management students showing the most inadequate knowledge levels followed by those enrolled in culinary nutrition, as shown in Table 5. The level of knowledge on nutrition among the students of the nutrition and dietetics program was predominantly adequate and satisfactory (73%). In addition, as shown in Table 6, the level of knowledge on nutrition was also associated with students’ eating habits. Nonetheless, eating habits were not associated with knowledge on cooking and techniques, BMI, or the year of study.

## 4. Discussion

BMI was used as an indicator of nutritional status in this study. No association was observed between BMI and eating habits as the majority of students reported having unhealthy eating habits, similar to the results of other studies [35]. On the other hand, a relationship between BMI and academic program type was observed. More than half of the students in the nutrition and dietetics program had a healthy BMI, compared to those in the culinary management program, in which the smallest proportion of students with a healthy BMI was observed, with almost half of the students classified as obese. As the great majority of students reported having inadequate eating habits, no relationship was observed between these two factors. Nevertheless, the highest proportion of unhealthy BMI was found in students with inadequate eating habits. Similarly, no correlation was observed between the level of knowledge on nutrition and BMI, indicating that knowledge alone does not necessarily promote adequate BMI [36]. Discerning the relationship between cooking skills and BMI is important, since the preparation of meals at home is increasingly promoted as a strategy to reduce the incidence of overweight and obesity [18,19] However, unlike in other studies [16,37], this relationship was not reflected in those enrolled in the culinary nutrition and culinary management programs, who showed adequate and satisfactory levels of knowledge on cooking methods and techniques, but inadequate levels of knowledge on nutrition and inadequate BMI values. These results indicate that the presence of culinary skills does not influence the maintenance of an adequate BMI in these students [21,38].

The majority of students reported having unhealthy eating habits, in agreement with previous studies [2,7,8]. In this study, a strong relationship was observed between eating habits and knowledge on nutrition. Similarly, eating habits were associated with the academic program type, indicating that a student’s nutrition-related knowledge, which was also associated with the academic program type, may contribute to the intake of a healthier diet. In this cohort, there were no first-year students in the nutrition and dietetics program, which may explain the greater level of knowledge on nutrition in this particular group than in the other two programs. Nonetheless, almost all the culinary management students had inadequate levels of knowledge on nutrition, unhealthier BMI values, and deficient eating habits.

Furthermore, this study showed that there are other factors inherent to students’ life that may have a stronger influence on eating habits. This is similar to Al-Qahtani et al.’s [39] findings, which presented the fact that nutritional education alone, although essential, is insufficient to influence eating habits since it does not address personal, behavioral, environmental, or economical barriers, as well as health conditions, stress, and genetic or hereditary elements [15,40].

Although this study makes several contributions to the existing literature, there are some limitations that should be considered. Firstly, validity assessments were not conducted for the survey, implying that none of the measured behaviors and conclusions can be confirmed. In addition, the sample was limited to students enrolled in study fields related to nutrition and culinary arts in the only educational institution in Puerto Rico that houses all three academic programs at a bachelor level, which limits generalizability of the results.

The majority of students presented inadequate eating habits, as established by the survey, which inhibited the possibility of clearly distinguishing additional factors that may influence eating habits. Likewise, the cross-sectional nature of the study limits the possibility of observing possible changes in students’ knowledge levels, eating habits, or nutritional status during their academic progression.

## 5. Conclusions

The eating habits of the university students were inadequate, revealing that enrollment in academic programs related to nutrition and culinary arts does not necessarily promote healthy eating habits. Nonetheless, statistically significant data showed that eating habits were related to students’ levels of knowledge on nutrition but not cooking methods and techniques. Students enrolled in the nutrition and dietetics program had higher levels of knowledge on nutrition but also presented inadequate eating habits, concluding that they presented the same barriers in relation to their eating habits as the culinary nutrition and culinary management students. In turn, the culinary management students showed knowledge levels that were superior to those observed among students enrolled in the other two programs in relation to cooking methods and techniques; however, they presented the same inadequate eating habits as the rest of the population.

In relation to the students’ nutritional status, it was found that there was an association between students’ BMI and academic program, evidencing that students in academic programs such as nutrition and dietetics may be more aware of their BMI. However, no correlation between students’ BMI and knowledge on nutrition, cooking methods, and techniques was found in this study. It was also determined that students with adequate and inadequate BMI values had inadequate eating habits, encouraging the need to investigate the different factors that influence BMI among university students. Furthermore, it is appropriate to mention that there were gaps between the knowledge and its application, and that other factors inherent to students’ university life may be present, the strength of which may surpass that of knowledge on appropriate nutritional practices.

## Figures and Tables

**Table 1 nutrients-12-01408-t001:** Sample characteristics of student profile are shown for three academic programs related to nutrition and culinary arts. BMI—body mass index.

Variable	University Career						*p*-Value
	Nutrition and Dietetics		Culinary Nutrition		Culinary Management		
	*n*	%	*n*	%	*n*	%	
**Gender**							0.121
Male	9	39.1	6	21.4	19	45.2	
Female	14	60.9	22	78.6	23	54.8	
**Age (Median, SD)**	24.09	6.24	27.75	8.77	22.24	4.24	0.020 *
**Marital Status**							0.091
Single	18	78.3	16	57.1	35	83.3	
Married/Cohabit	5	21.7	10	35.7	6	14.3	
Divorced	-	-	1	3.6	0	-	
Widowed	-	-	1	3.6	1	2.4	
**Residence**							<0.001 *
Family residence under caretaker	18	78.3	10	35.7	36	85.7	
Caretaker of family residence	3	13.0	16	57.1	4	9.5	
Student housing	2	8.7	2	7.1	2	4.3	
**Income**							0.039 *
Low	14	60.9	22	78.6	37	88.1	
Medium	9	39.1	6	21.4	5	11.9	
**Study and work**							0.018 *
Full-time student	9	24.3	10	35.7	18	42.9	
Full-time student and work	12	39.1	5	18.9	10	23.8	
Part-time student	2	9.7	13	46.4	14	33.3	
**Cooks in home**							0.103
Yes	21	91.3	28	100	36	85.7	
No	2	8.7	-	-	6	14.3	
**Plans meals before grocery shopping**							0.290
Yes	11	47.8	15	53.6	15	35.7	
No	9	39.1	12	42.9	25	59.5	
Does not shop for groceries	3	13.0	1	3.6	2	4.8	
**Knows daily intake of calories**							0.011 *
Yes	21	91.3	13	46.4	27	64.3	
No	2	8.7	15	53.6	15	35.7	
**Type of diet**							0.372
Omnivores	15	65.2	21	75.0	34	80.9	
Other	8	34.8	7	25.0	8	19.1	
**Physical Activity**							0.740
Yes	17	73.9	18	64.3	30	71.4	
No	6	26.1	10	35.7	12	28.6	
**Nutritional status (BMI)**							0.014 *
Underweight	3	13.0	1	3.6	1	3.4	
Normal weight	13	56.5	14	50.0	11	26.2	
Overweight	5	21.7	8	28.6	12	28.6	
Obese	2	8.7	5	17.9	18	42.9	
**Year of study**							0.280
Freshmen	-	-	5	17.9	7	16.7	
Sophomore	7	30.4	6	21.4	10	23.8	
Junior	6	26.1	8	28.6	15	51.7	
Senior	10	43.5	9	32.1	10	23.8	

* *p*-value less than 0.05.

**Table 2 nutrients-12-01408-t002:** Frequency and percentage of students in academic programs related to nutrition and culinary arts with correct eating habit according to dietary guidelines.

Eating Habit According to Recommended Guidelines	Students with Adequate Eating Habits	
	*n*	%
Fat (butter, margarine, oils, and other fat)	83	89.3
Low-sugar sodas (diet soda)	80	86.0
Boiled meals	80	86.0
Eats lunch every day	73	78.5
Eats candies or sweets	69	74.2
Eats food made from scratch	69	74.2
Whole-grain flour	69	74.2
Considers the list of ingredients of the products to consume	68	73.1
Eats charcuterie (sausage, cured meats)	45	48.4
Whole milk/reduced fat milk	45	48.4
Drinks water	44	47.3
Portion control	40	43
Daily intake of 2–3 snacks	32	34.4
Eats fruits (fresh, canned, or frozen)	27	29.0
Eats salads or vegetables (fresh, canned, or frozen)	19	20.4
Eats fried food	19	20.4

**Table 3 nutrients-12-01408-t003:** Frequency and percentage of students in academic programs related to nutrition and culinary arts with correct knowledge on questions related to nutrition.

Premise	Students with Correct Knowledge	
	*n*	%
Fiber content of legumes	46	49.5
Amount of fruits and vegetable servings per day	46	49.5
Foods in balanced breakfast	45	48.4
Recommended dairy products	45	48.4
Main source of calcium	45	48.4
Recommended water intake per day	39	41.9
Contribution of breakfast on daily diet	36	38.7
Recommended strategies to lose weight	33	35.5
Calories represented in nutritional label: per package	30	32.3
Recommended sugar and sodium content in nutritional label	25	26.9
Definition of overweight	21	22.6

**Table 4 nutrients-12-01408-t004:** Frequency and percentage of students in academic programs related to nutrition and culinary art with correct knowledge on cooking methods and techniques, and adequate application of culinary competencies and knowledge.

Variables	Students with Correct Knowledge or Adequate Competencies	
	*n*	%
**Knowledge on cooking methods and techniques**		
Definition of boiling	89	95.7
Definition of microwave cooking	85	91.4
Using spices to add flavor to food without using salt or fat	84	90.3
Cooking technique for the retention of chlorophyll.	32	34.4
Sous-vide reduces food mass	31	33.3
Using low sodium salt to add flavor to food without using additional salt or fat	16	17.2
**Culinary competencies**		
Food preservation techniques	64	68.8
Cooking carbohydrates	89	95.7
Reading recipes	82	88.1
Using leftovers to prepare new meals	51	54.8
Cooking technique: boil	89	95.7
Cooking technique: steam	89	95.7
Cooking technique: sous-vide	43	46.2
Cooking technique: baking	84	90.3
Cooking technique: smoked	54	58.1
Cooking method: fat	56	60.2
**Application of knowledge**		
Prepares meal with more than three ingredients	78	83.8
Visualize food and its plating before cooking	76	81.7
Use of measuring equipment while cooking from scratch	51	54.8
Preparation of grocery list and plan what to eat daily or during the week	46	49.5

**Table 5 nutrients-12-01408-t005:** Frequency and percentage of students in three nutrition and culinary arts programs and the relationship with eating habits, knowledge on nutrition, and knowledge on cooking methods and techniques.

Variable	Academic Programs						***p*-Value**
	Nutrition and Dietetics		Culinary Nutrition		Culinary Management		
	*n*	%	*n*	%	*n*	%	
**Eating habits**							0.010 *
Adequate	3	13	-	-	1	2.4	
Satisfactory	2	8.7	6	21.4	1	2.4	
Inadequate	18	78.3	22	78.6	40	95.2	
**Knowledge on nutrition**							<0.001 *
Adequate	10	43.5	2	7.1	1	2.4	
Satisfactory	7	30.4	8	28.6	1	2.4	
Inadequate	6	26.1	18	64.3	40	95.2	
**Knowledge on cooking methods and techniques**							0.346
Adequate	4	17.4	12	42.9	14	33.3	
Satisfactory	6	26.1	6	21.4	12	28.6	
Inadequate	13	56.5	10	35.7	16	38.1	

* *p*-value less than 0.05.

**Table 6 nutrients-12-01408-t006:** Frequency and percentage of students’ eating habits in three nutrition and culinary arts programs and the relationship with knowledge on nutrition and knowledge on cooking methods and techniques, nutritional status, and classification.

Variable	Eating habits						*p*-Value
	Adequate		Satisfactory		Inadequate		
	*n*	%	*n*	%	*n*	%	
**Knowledge on nutrition**							0.002 *
Adequate	3	75	3	33.3	7	8.7	
Satisfactory	0	-	3	33.3	13	16.2	
Inadequate	1	25	3	33.3	60	75	
**Knowledge on cooking methods and techniques**							0.608
Adequate	2	50	4	44.4	24	30	
Satisfactory	1	25	3	33.3	20	25	
Inadequate	1	25	2	22.2	36	45	
**BMI**							0.165
Underweight	0	-	2	22.2	3	3.5	
Normal weight	2	50	4	44.4	32	40	
Overweight	1	25	3	33.3	21	26.2	
Obese	1	25	0	-	24	30	
**Year of study**							0.190
Freshmen	0	-	0	-	12	15	
Sophomore	0	-	3	33.3	20	25	
Junior	1	25	1	11.1	27	33.7	
Senior	3	75	5	55.5	21	26.2	

* *p*-value less than 0.05.

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
