# Peer review of "Eating Habits Associated with Nutrition-Related Knowledge among University Students Enrolled in Academic Programs Related to Nutrition and Culinary Arts in Puerto Rico"

_nutrients, 2020, doi:10.3390/nu12051408_

Round 1

Reviewer 1 Report

Those studies are important because students of nutrition-related fields should have knowledge of how to eat properly. In addition, research confirms the sense of conducting nutrition programs. The results are justified by experimental observations. In general the manuscript is clear and properly assessed and adds information to the area of interest. The authors thoroughly described the research methodology and presented and discussed the results. The discussion undertaken is sufficient. The authors are aware that the study was conducted on a small group of students. This creates limitations on wider inference. However, the work is interesting and can be an introduction to further research.

L87: should be “Rizo-Baeza”

L373; L407-408; L420-421 - remove becket

Author Response

Dear Reviewer 1

Thanks for your comments and recommendations in regards to Manuscript ID: nutrients-779380. I have some difficulty in understanding what do you mean by “Removing Becket”. Furthermore, I have implemented a table that will facilitate the identification of the changes made based on my understanding to your comments.

Thanks

Christian Rivera Medina

Line

Reviewer comment

Original text

Implemented changes

87

should be “Rizo-Baeza”

Rizo-baeza

Rizo-Baeza

373

Remove Becket

Mihalopoulos NL, Auinger P, Klein JD. The Freshman 15: Is it real? J Am Coll Health. 2008;56(5):531-534.

Mihalopoulos NL, Auinger P, Klein JD. The Freshman 15: Is it real? J Am Coll Health. 2008;56.5.531-534

407

Remove Becket

Prev Med Rep. 2016;4:23-28.

Not aware of what change to be made.

408

Remove Becket

Reicks M, Trofholz AC, Stang JS, Laska MN. Impact of cooking and home food preparation interventions

Not aware of what change to be made.

420

Remove Becket

Tamayo AP, Cubero J, Constantino J, Macrás R. Previous knowledge in nutrition of a group of students of

Tamayo AP, Cubero J, Constantino J, Macras R. Previous knowledge in nutrition of a group of students of

421

Remove Becket

secondary education of a penitentiary Spanish center.

Not aware of what change to be made.

Reviewer 2 Report

The manuscript presents the eating habits of university students participating in academic programs related to nutrition and culinary arts in Puerto Rico. The questionnaire consisted four sections: student profile, eating habits, knowledge of nutrition, and knowledge of cooking methods and techniques. The results obtained indicate that most students had inadequate eating habits, knowledge in nutrition and BMI values. In addition, the results show that there are other factors associated with the university life of students that may have a greater impact on eating habits than knowledge of appropriate nutritional practice.

The number of students participating in the research was 93. Please answer, what was the reason that such a small number of students were participating in the study? If non-probabilistic sampling was used.

Abstract should contain maximum 200 words. Please shorten this paragraph. The sentence regarding statistical analysis (line 26-28) should be removed. Line 28-29: “An inadequate body mass index (BMI) was observed in 59% of the sample”. It’s better to use “students” or “population” than “sample”.

Line 38: I suggest add to keywords: eating habits, nutritional requirements

Line 206-207: The values presented in the tables do not include standard deviations?

Line 214-217: Write, what was the reason that such a small number of students were participating in the study?

Line 257: “Table 3 shows the questions that were answered incorrectly by the majority of students”. This sentence should be corrected (. Table 3 shows correct knowledge in questions related to nutrition.

Line 269: … 61.4% (SD=18)… ?! Is it about value 61.4 ± 18.0% ?!

Table 6 should be prepared in the same way (remove vertical lines) as Table 1-5.

Discussion section should be extended with literature confirming the sentences: line 35-37 and line 349-351. Discuss additional (other) factors that may effect on eating habits. In addition, it is important to developing discussion section about the influence of low income on eating habits. 78.5% of the students were classified as having a low income (line 227-228)!

Line 313-314: …… agreement with previous students…. “students”?! It should be improved on “studies”.

References in the sections 1-4 in the manuscript should be numbered in square brackets, e.g., [1] or [2,3], or [4–6].

Author Response

Dear Reviewer 2

Thanks for your comments and recommendations in regards to Manuscript ID: nutrients-779380. Furthermore, I have implemented a table that will facilitate the identification of the changes made based on my understanding to your comments.

Thanks

Christian Rivera Medina

Original Line

Reviewer comment

Original text

New line

Implemented changes

Abstract

Abstract should contain maximum 200 words. Please shorten this paragraph.

Background: University students frequently develop unhealthy eating habits. However, it is unknown if students enrolled in academic programs related to nutrition and culinary arts have healthier eating habits. Objective: To evaluate the relationship of the eating habits and nutritional status of university students enrolled in academic programs related to nutrition and culinary arts with their knowledge in nutrition as well as cooking methods and techniques. Design: This descriptive cross-sectional study was conducted in the spring of 2019, during which time students completed a survey that measured their eating habits, and their knowledge in nutrition as well as cooking methods and techniques. Anthropometric measurement data were also collected for nutritional status estimation. Participants/setting: The non-probabilistic convenience sample comprised 93 university students pursuing their Bachelor’s degree in Nutrition and Dietetics, Culinary Nutrition and Culinary Management in Universidad Ana G. Mendez, Puerto Rico. Statistical analyses performed: The statistical analysis included summary measures. Relationships between variables were measured using Chi-square and Fisher’s exact tests, and statistical significance was set at p<0.05. Results: An inadequate body mass index (BMI) was observed in 59% of the sample. Eating habits, knowledge in nutrition, and knowledge in cooking methods and techniques were inadequate in 86%, 68%, and 41% of the population, respectively. Eating habits were associated with knowledge in nutrition and the type of academic program enrolled in, but not with knowledge in cooking methods and techniques. Conclusions: Most of the students reported having inadequate eating habits and BMI values. The Nutrition and Dietetics students had the best knowledge levels pertaining to nutrition compared to the Culinary Management students, a majority of whom had inadequate knowledge levels. Our results demonstrate that there are other factors inherent to students’ university life that may have a stronger influence on eating habits than knowledge in appropriate nutritional practice.

Abstract

15-29

200-word count.

University students frequently develop unhealthy eating habits. However, it is unknown if students enrolled in academic programs related to nutrition and culinary arts have healthier eating habits. Evaluated the relationship of eating habits and nutritional status of students in academic programs with knowledge in nutrition as well as cooking methods and techniques. Descriptive cross-sectional study conducted on spring of 2019, completed a survey measuring eating habits, knowledge in nutrition as well as cooking methods and techniques. Anthropometric measurement was collected for nutritional status estimation. Non-probabilistic convenience sample comprised 93 students pursuing degrees at Universidad Ana G. Mendez, Puerto Rico. Inadequate BMI was observed in 59% of the students. Eating habits, knowledge in nutrition, and knowledge in cooking methods and techniques were inadequate in 86%, 68%, and 41% of the population, respectively. Eating habits were associated with knowledge in nutrition and academic program, but not with knowledge in cooking methods and techniques. Most students reported having inadequate eating habits and BMI. Nutrition and Dietetics students had best knowledge in nutrition compared to Culinary Management students, a majority of whom had inadequate knowledge. Concluding that there are other factors inherent to students’ life that may have a stronger influence on eating habits.

26 – 28

The sentence regarding statistical analysis (line 26-28) should be removed.

Statistical analyses performed: The statistical analysis included summary measures. Relationships between variables were measured using Chi-square and Fisher’s exact tests, and statistical significance was set at p<0.05.

Sentence deleted

29 – 29

An inadequate body mass index (BMI) was observed in 59% of the sample”. It’s better to use “students” or “population” than “sample”.

An inadequate body mass index (BMI) was observed in 59% of the sample.

22-23

An inadequate body mass index (BMI) was observed in 59% of the students.

38

I suggest add to keywords: eating habits, nutritional requirements

university; student; nutrition; knowledge; culinary arts

30

university; student; nutrition; knowledge; culinary arts, eating habits, nutritional requirements

206-207

The values presented in the tables do not include standard deviations?

Age (Median, IQR)

Table 1

289

Age (Median, SD)

214-217

Write, what was the reason that such a small number of students were participating in the study?

The total population comprised 93 students.

273-274

At the time this investigation took place their where only a total of 93 students enrolled dispersed with in the three academic programs that complied with inclusion criteria.

262

“Table 3 shows the questions that were answered incorrectly by the majority of students”. This sentence should be corrected (. Table 3 shows correct knowledge in questions related to nutrition.

Table 3 shows the questions that were answered incorrectly by the majority of students.

330

Table 3 shows correct knowledge in questions related to nutrition by the majority of students.

269

61.4% (SD=18)… ?! Is it about value 61.4 ± 18.0%?!

61.4% (SD=18)

343

61.4±18%

Table 6

Table 6 should be prepared in the same way (remove vertical lines) as Table 1-5.

Table 6 3

64

Vertical lines removed

35 -37

349 – 351

Discussion section should be extended with literature confirming the sentences: Discuss additional (other) factors that may effect on eating habits.

35-37: Our results demonstrate that there are other factors inherent to students’ university life that may have a stronger influence on eating habits than knowledge in appropriate nutritional practice.

349-351:

Furthermore, it is appropriate to mention that there were gaps between the knowledge and its application, and that other factors inherent to students’ university life may be present, the strength of which may surpass that of knowledge in appropriate nutritional practices.

398- 402

Furthermore, this study showed that there are other factors inherent to students’ life that may have a stronger influence on eating habits. Similar to Al-Qahtani et al. [34] findings, in which his study presented the fact that nutritional education alone, although essential, is insufficient to influence eating habits since it does not address personal, behavioral, environmental or economical barriers, as well as health conditions, stress, genetic or hereditary elements [35,36].

227-228

In addition, it is important to developing discussion section about the influence of low income on eating habits. 78.5% of the students were classified as having a low income

313- 314

…… agreement with previous students…. “students”?! It should be improved on “studies”

The majority of students reported having unhealthy eating habits, in agreement with previous students.

390

The majority of students reported having unhealthy eating habits, in agreement with previous studies.

References

In the sections 1-4 in the manuscript should be numbered in square brackets, e.g., [1] or [2,3], or [4–6].

All references have been updated to meet specifications with in the manuscript.
